



# A decade of dissolved sulphur compounds measurements in the southwestern Baltic Sea

Yanan Zhao[1], Cathleen Schlundt[1], Dennis Booge[1] and Hermann W. Bange[1]

[1]GEOMAR Helmholtz Centre for Ocean Research Kiel, Düsternbrooker Weg 20, 24105 Kiel, Germany

*Correspondence to*: Yanan Zhao (yzhao@geomar.de)

**Abstract.** Dissolved sulphur compounds were measured at the Boknis Eck Time-Series Station (BE, Eckernförde Bay, SW Baltic Sea) during the period February 2009–December 2018. Our results show considerable interannual and seasonal variabilities in the mixed layer concentrations of dimethyl sulphide (DMS), total dimethylsulphoniopropionate ($DMSP_t$) and total dimethyl sulphoxide ($DMSO_t$). Positive correlations were found between particulate DMSP ($DMSP_p$) and particulate

DMSO ($DMSO_p$) as well as $DMSP_t$ and $DMSO_t$ in the mixed layer, suggesting a similar source for both compounds. The decreasing long-term trends, observed for $DMSP_t$ and DMS in the mixed layer, were linked to the concurrent trend of the sum of 19′-hexanoyloxyfucoxanthin and 19′-butanoyloxy fucoxanthin, which are the marker pigments of prymnesiophyceae and chrysophytes, respectively. Major Baltic Inflow (MBI) events influenced the distribution of sulphur compounds due to phytoplankton community changes, and sediment might be a potential source for DMS in the bottom layer during seasonal

hypoxia/anoxia at BE. A modified algorithm based on the phytoplankton pigments well reproduces the $DMSP_p$:Chl *a* ratios during this study and could be used to estimate future surface (5 m) $DMSP_p$ concentrations at BE.

## 1 Introduction

Dimethyl sulphide (DMS) plays an essential role in the sulphur cycle of the Earth's atmosphere (Lovelock et al., 1972): DMS released from the ocean surface may affect the Earth's climate by forming atmospheric sulphate aerosols which, in turn, can

backscatter solar radiation and possibly act as cloud condensation nuclei that form clouds. Both processes have a cooling effect on the atmosphere (Charlson et al., 1987; Vogt and Liss, 2009; Wang et al., 2015). However, the global significance of this DMS-driven ocean/climate feedback mechanism remains elusive (Quinn and Bates, 2011; Green and Hatton, 2014; Wang et al., 2018).

The production and consumption of DMS are affected by complex and interacting processes regulated by environmental and

biogeochemical factors (Stefels et al., 2007; Vogt and Liss, 2009; Asher et al., 2011 ). Marine-derived DMS is produced from its major precursor dimethylsulphoniopropionate (DMSP) mainly by enzymatic cleavage of DMSP into DMS and acrylate (Curson et al., 2011). However, this pathway is only of minor importance for DMSP loss (generally accounting for 10 %), since most of the DMSP is directly consumed by phytoplankton and bacteria (Vila-Costa et al., 2006; Moran et al., 2012). The



primary loss processes of dissolved DMS include (i) microbial consumption, (ii) photooxidation, (iii) air-sea gas exchange and

(iv) vertical export by mixing (Simo, 2004).

DMSP is mainly produced in the cells of algae and bacteria as a response to multiple environmental stressors (Simo, 2004; Stefels et al., 2007; Schäfer et al., 2009; Alcolombri et al., 2015; Curson et al., 2017). Certain phytoplankton species, such as dinophyceae and haptophyceae, show high DMSP production rates while diatoms are low DMSP producers (Keller et al., 1989; Kirst et al., 1991). Intracellular DMSP is involved in a variety of physiological functions such as osmoregulation

(Vairavamurthy et al., 1985), cryoprotection (Kirst et al., 1991; Lee and De Mora, 1999), antioxidation (Sunda et al., 2002; Simó and Vila-Costa, 2006), methyl donation (Kiene et al., 2000), grazing deterrence (Wolfe et al., 2002) or overflow mechanism during nitrogen limited conditions (Stefels, 2000). Therefore, DMSP production in phytoplankton is also dependent on the ambient environmental conditions mentioned above. DMSP is released by phytoplankton into the marine environment due to senescence, zooplankton grazing, and virus infections (Stefels, 2000; Stefels et al., 2007).

Although DMSO is as ubiquitous as DMSP in surface seawater, its formation and consumption pathways are still poorly understood (Green et al., 2011; Hatton et al., 2012). DMSO mainly originates from the photochemical and bacterial oxidation of DMS, as well as direct synthesis in marine algae cells (Lee and De Mora, 1999; Lee et al., 1999). The sinks of DMSO include bacterial consumption and reduction to DMS (Hatton et al., 2004). Only recently it was found by Thume et al. (2018) that dimethysulphoxonium propionate (DMSOP) is an intermediate when forming DMSO from DMSP and this alternative

DMSO production pathway circumvents DMS production. DMSO possesses similar intracellular functions as DMSP in algae cells (Simo et al., 1998; Sunda et al., 2002).

Long-term observations are a valuable tool to monitor and decipher short- and long-term trends in oceanic environments (Ducklow et al., 2009). To this end, several time-series studies of DMS from different open ocean and coastal sites, such as the North Sea, the Atlantic Ocean and the Indian Ocean, have been conducted during the past years (see e.g. (Turner et al.,

1996; Dacey et al., 1998; Shenoy and Patil, 2003; Vila-Costa et al., 2008; Dixon et al., 2020). However, the distributions and cycling of sulphur compounds in the Baltic Sea are still largely unknown, and only a few studies of DMS were carried out in the Baltic Sea (Leck et al., 1990; Leck and Rodhe, 1991; Orlikowska and Schulz-Bull, 2009). Here we present a dataset of long-term observations of DMS, DMSP, and DMSO as well as biotic and abiotic parameters from the Boknis Eck Time-Series Station (BE) located in the Eckernförde Bay (southwestern Baltic Sea). To our knowledge, this is the longest and most

comprehensive time-series measurement of sulphur compounds so far. The overarching objectives of this study are to decipher (i) seasonal and long-term trends of the sulphur compounds, (ii) the influence of extreme events such as Major Baltic Inflow (MBI) and low oxygen events on the sulphur cycling, (iii) how the phytoplankton composition influences the seasonal distributions of the sulphur compounds.





## 2 Sampling area

Sampling was performed at BE (www.bokniseck.de), whose site is located at the entrance of the Eckernförde Bay (54° 31.2' N, 10° 02.5' E; Fig. 1) in the southwestern Baltic Sea. The BE sampling site has a water depth of 28 m. Monthly sampling at BE started in 1957, making this station one of the longest-operating marine time-series stations worldwide (Lennartz et al., 2014). Riverine inputs are negligible for the Eckernförde Bay which is dominated by the inflow of North Sea water through the Kattegat and the Great Belt. Seasonal stratification at BE is caused by steep density gradients and usually lasts from March

to October with a mixed layer depth of 10–15 m (Hoppe et al., 2013; Lennartz et al., 2014). During the stratification period, vertical mixing is restricted and decomposition of organic material by bacteria in the deep layer causes pronounced hypoxia and sporadically anoxia or sulphidic events (Hansen et al., 1999; Lennartz et al., 2014). The main phytoplankton blooms generally occur in spring (February–March). Minor blooms are sporadically in summer (July–August) and always in autumn (September–November) (Smetacek et al., 1984; Smetacek, 1985; Bange et al., 2010). Lennartz et al. (2014) reported an

increasingly warming trend of 0.02 °C yr$^{-1}$ (in 1 m and 25 m) at BE for the period from 1957 till 2013. Nutrients concentrations increased until the 1980s in the Baltic Sea, as a result of agricultural over-fertilisation, washing-off and transport via rain and rivers into the Baltic Sea. The nutrient concentration started to decline due to measures which successfully reduced anthropogenic caused marine eutrophication in the Baltic Sea (HELCOM, 2018). However, low-oxygen events (hypoxia or anoxia) occur more frequently within the last decades in the Baltic Sea and so at BE (Lennartz et al., 2014). Probably, climate

warming enhances bacterial activities and respiration (Hoppe et al., 2013) and extends the period of stratification (Liblik and Lips, 2019). This overrides the effect of decreasing nutrient inputs in the last decades (Lennartz et al., 2014). Overall, the location of BE is ideal for studying the cycling of sulphur compounds such as DMS, DMSP and DMSO in a productive coastal ecosystem with strong open ocean influences, which is affected by pronounced changes in salinity and oxygen.

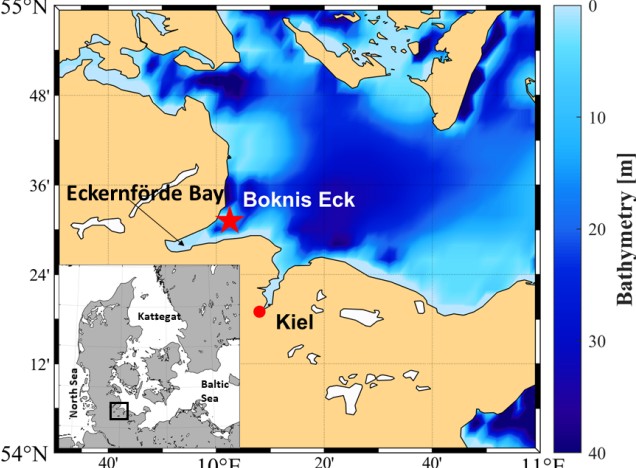


**Figure 1: Location of the Boknis Eck Time-Series Station near the entrance of Eckernförde Bay in the southwestern Baltic Sea.**





# 3 Material and methods

## 3.1 Sulphur compounds analysis

Monthly sampling of sulphur compounds at BE started in February 2009. Samples were collected bubble-free in 250 mL brown

glass bottles. The samples were analysed as soon as possible after returning to GEOMAR's laboratory, usually within a few

hours after sampling. Back in the lab, out of the 250 mL water sample, three subsamples (10 mL) were immediately taken and

gently filtered through a glass fibre filter (GF/F; Whatman; 0.7 µm) attached to a syringe for DMS and dissolved DMSP

($DMSP_d$) analysis. We used a purge and trap technique attached to a gas chromatograph equipped with a flame photometric

detector (GC-FPD) to measure sulphur compounds as described in Zindler et al. (2012). After DMS was measured, sodium

hydroxide (NaOH; Carl Roth) was added to the subsamples to convert DMSP into DMS. The conversion was allowed to take

place at least overnight before analysis of $DMSP_d$. Total DMSP ($DMSP_t$) was measured from the unfiltered alkaline sub-

samples, and particulate DMSP ($DMSP_p$) concentrations were calculated by subtracting measured DMS and $DMSP_d$

concentrations from measured $DMSP_t$ concentrations. Dissolved DMSO ($DMSO_d$) and total DMSO ($DMSO_t$) samples were

measured from the same samples of $DMSP_d$ and $DMSP_t$ measurements by adding cobalt-dosed sodium borohydride ($NaBH_4$;

Sigma-Aldrich) right after DMSP analysis to reduce DMSO to DMS. Particulate DMSO ($DMSO_p$) concentrations were

calculated by subtracting measured $DMSO_d$ concentrations from measured $DMSO_t$ concentrations. Calibrations were

conducted every measurement day. The mean relative analytical errors for the individual sulphur compounds were generally

$\leq 20$ %. An overview of the methods used for determining oceanographic parameters such as water temperature, salinity,

dissolved $O_2$ and dissolved nutrients at BE can be found in Lennartz et al. (2014).

## 3.2 Phytoplankton analysis

Pigment samples were collected simultaneously with sulphur compounds samples at BE. After returning to the lab, 2 L of

seawater was filtered through 0.7 µm GF/F glass fibre filters with a pressure of less than 200 mbar to avoid cell breaking. After

filtration, the filters were folded and stored in 2 mL microcentrifuge tubes (Eppendorf cups) at -80 °C for later analysis.

Phytoplankton pigment concentrations from April 2009–December 2011 were analysed using a High Performance Liquid

Chromatography (HPLC, Waters Pumpe 600 Scan Fluorometer 474 Photodiodenarray 2996 Autosampler 717) technique.

50 µL of an internal standard (canthaxanthin) and 2 mL 100 % acetone were added and the pigments were extracted by

homogenisation with glass beads in a cell mill (Bühler). Samples were centrifuged and the supernatant was filtered through

0.2 µm PTFE filters (VWR International). Just prior to analysis, the sample was premixed with 1 mol $L^{-1}$ ammonium acetate

solution in the ratio 1:1 (v/v) in the autosampler and injected onto the HPLC system. The pigments were analysed by reverse-

phase HPLC, using a VARIAN Microsorb-MV3 C8 column (4.6 × 100 mm) and HPLC-grade solvents (Baker). The gradient

was modified after Barlow et al. (1997). Eluting pigments were detected by absorbance (440 nm). From 2012 on, just prior to

analysis, the sample was premixed with 28 mM Tetrabutylammoniumacetate solution in the ratio 1:1 (v/v) in the autosampler

and injected onto the HPLC system. The pigments were analysed by reverse-phase HPLC, using an Eclipse XDB-C8 column



(4.6 × 150 mm) and HPLC-grade solvents (Baker). The gradient was modified after Van Heukelem and Thomas (2001), and

eluting pigments were detected by absorbance (440 nm). Both methods showed a good agreement in a pigments analysis, thus data are comparable before and after 2012.

The taxonomic structure of phytoplankton communities was derived from photosynthetic pigment ratios using the CHEMTAX® program (Mackey et al., 1996), applying the input matrix of Eker-Develi et al. (2008) and Schluter et al. (2000). The phytoplankton group composition included diatoms, dinoflagellates, cryptophytes, chrysophytes, chlorophytes,

prymnesiophyceae and cyanobacteria.

**3.3 Mixed layer depth (MLD)**

With 28 m water depth, the BE station is a shallow coastal site. Therefore, a density-based criterion for calculating the MLD is the best approach. In order to define the MLD, we used the squared buoyancy frequency ($N^2$), also called stability frequency, which was calculated following Eq. (1):

$$N^2 = \frac{g}{\rho} \frac{d\rho}{dz},\tag{1}$$

by using the water density ($\rho$), the water depth (z) and the gravity (g). After calculating $N^2$ for all depth profiles of this dataset, the MLD was defined as the minimum depth below 4 m where the criterion of $N^2 \geq 10^{-3}$ s$^{-2}$ was true. This $N^2$-value is low enough to detect a barrier where mixing is mainly suppressed but also high enough not to account for a diurnal surface warm layer as the MLD, as the MLD is applied for the whole month in which the individual cruises took place.

**4. Results and discussion**

**4.1 Overview**

**4.1.1 Environmental setting of BE during 2009-2018**

The water temperature varied between 0.42 and 22.15 °C (Fig. 2a), with a maximum usually in August (14.56 °C; Fig. 2b) and a minimum between February (2.88 °C; Fig. 2b) and March (2.51 °C; Fig. 2b). The highest water temperature of 22.15 °C was

measured in 1 m water depth in August 2018, which was the warmest summer recorded for the Baltic Sea since 1948 (Siegel and Gerth, 2018) and also the second warmest summer in Germany since 1981 (Zscheischler and Fischer, 2020). The lowest water temperature of 0.42 °C was measured in 15 m water depth in March 2010. In general, the temperature of the water column at BE increased with 0.02 °C yr$^{-1}$ during 1957–2013 due to global warming (Lennartz et al., 2014). The salinity in the bottom layer (25 m) ranged from 13.65 to 25.66, with the highest salinity measured in December 2013 (Fig. 2c). In general,

the salinity in 25 m water depth reached its maximum in September (23.09; Fig. 2d) after the stratification period and its minimum in April (19.64; Fig. 2d) when the water column is well-ventilated by wind-driven mixing events. The bottom salinity showed strong fluctuations which are caused due to the inflow of saline water originating from the North Sea (Lennartz et al., 2014). For instance, in December 2014, a MBI event of high saline and oxygenated North Sea water occurred after a 10-year





stagnation since 2003, as the third-strongest event ever recorded (Fig. 2c, marked with the black arrow) (Mohrholz, 2018).

Occasionally, the break-up of the late summer/autumn stratification was caused by upwelling events induced by strong winds, leading to the uniform distribution of salinity in the entire water column (e.g. in September 2017).

**Figure 2: Monthly and mean seasonal distributions of temperature (a, b), salinity (c, d), dissolved $O_2$ (e, f), phosphate (g, h) and dissolved inorganic nitrogen (i, j) at BE during 2009–2018. Please note that in the left panels, the blank areas are due to data gaps**
**caused by cancellations of the research cruises and the dashed lines indicate January of each year. Black dots and the black line (c and d) indicate monthly and mean seasonal distributions of the mixed layer depth, respectively. The black arrows (c) indicate the Major Baltic Inflow (MBI) events in November 2010 and December 2014, respectively. Hypoxia in e and f is defined as where dissolved $O_2$ concentrations were below 62.5 µmol L$^{-1}$ (i.e. 2 mg L$^{-1}$), according to Vaquer-Sunyer and Duarte (2008). The colour coding in g and i is shown on a natural logarithmic scale.**

Dissolved oxygen concentrations varied significantly from 0 to 479 µmol L$^{-1}$ (Fig. 2e), with seasonal hypoxic or anoxic events prevailing in the bottom layer (~ 20–25 m) in autumn at BE (Fig. 2f). Dissolved phosphate and total inorganic nitrogen



(DIN, the sum of nitrate, nitrite and ammonium) concentrations generally displayed regular seasonal variabilities, with higher concentrations in the upper layers during the winter months (December–February) and in the bottom layer during autumn months (September–November) (Fig. 2, g–j). The seasonal variability of chlorophyll a (Chl $a$) concentrations was generally in line with the annual phytoplankton succession at BE previously reported by Smetacek (1985), which is characterised by diatom blooms in spring, minor blooms in summer, dinoflagellate blooms in autumn and a dormancy phase in winter (Fig. 3a and 3b). During our study, autumn blooms at BE occasionally extended to December, which might have been a result of a longer growing season at higher temperatures in response to climate change (Wasmund et al., 2011). The highest Chl $a$ concentration (12.4 µg L$^{-1}$) was measured in the surface layer (1 m) in October 2017, accompanied by dinoflagellates dominating the autumn bloom.

### 4.1.2 Sulphur compounds

DMSP$_p$ concentrations were up to 103.5 nmol L$^{-1}$ with an average of 9.2 ± 13.3 nmol L$^{-1}$ in the water column, and DMSP$_d$ concentrations reached up to 42.7 nmol L$^{-1}$ with an average of 3.0 ± 4.1 nmol L$^{-1}$. The highest concentration of DMSP$_p$ was measured in 15 m depth in April 2015. Generally, the seasonal and spatial patterns of DMSP$_p$ and DMSP$_d$ followed that of Chl $a$, which was enhanced in spring (February–April) and autumn (September–October) in the upper layer (~ 1–15 m) and decreased with increasing depth (Fig. 3, a–f). The overall mean ratio of DMSP$_p$:DMSP$_d$ was 4.5 ± 8.5, indicating that DMSP$_p$ was generally dominant in the DMSP pool at BE. This is in line with the results reported by Speeckaert et al. (2018) from the coastal areas in the southern North Sea. DMSO$_p$ concentrations were up to 208.4 nmol L$^{-1}$ with an average of 11.3 ± 20.7 nmol L$^{-1}$ in the water column. DMSO$_d$ concentrations were up to 70.3 nmol L$^{-1}$ with an average of 7.9 ± 8.2 nmol L$^{-1}$. The highest DMSO$_p$ concentration was measured in 1m depth in the same sampling month as DMSP$_p$. The seasonal and spatial distributions of DMSO$_p$ and DMSO$_d$ were similar to DMSP (Fig3. I–l). The mean ratio of DMSO$_p$:DMSO$_d$ was 1.7 ± 2.4, suggesting less dominant of the DMSO$_p$ to DMSO$_d$ in contrast to DMSP. Overall, our study is consistent with the results reported in Hatton and Wilson (2007) who reported that DMSP$_d$ was very low compared to DMSP$_p$ while DMSO$_d$ could exceed the sum of DMS and DMSP$_d$ concentrations in the seawater. Additionally, significant correlations between DMSP$_p$ and DMSO$_p$ as well as DMSP$_t$ and DMSO$_t$ (Table 1) in this study, confirming previous studies that both compounds might share the same source in the seawater and they are subject to close cycling of production and consumption where the composition of the planktonic community, play a prominent role (Simo et al., 1997; Zindler et al., 2013).

The overall mean DMS concentration was 1.3 ± 1.8 nmol L$^{-1}$ in the water column, with the highest concentration of 20.5 nmol L$^{-1}$ measured in 1 m depth in April 2015. The mean concentration of DMS in the mixed layer was 1.7 ± 2.0 nmol L$^{-1}$, which is slightly lower compared to the mean DMS concentration of 2.7 ± 2.0 nmol L$^{-1}$ for the Baltic Sea (53°N–66°N, 10°E–30°E) retrieved from the global Surface Seawater DMS database (http://saga.pmel.noaa.gov/dms), including DMS data from Leck et al. (1990) and Leck and Rodhe (1991). DMS concentrations measured at the entrance of Himmerfjärden Fjord (western Baltic Sea) from January 1987 to June 1988 ranged from 0.1 to 6.3 nmol L$^{-1}$ with an average of 1.5 ± 1.3 nmol L$^{-1}$ (Leck et al., 1990), which is in line with our study. Surface DMS was also measured in the Baltic Sea and the Kattegat/Skagerrak (the





connection between the Baltic Sea and the North Sea) with $1.3 \pm 0.8$ and $2.4 \pm 0.9$ nmol $L^{-1}$ in July 1988, respectively (Leck and Rodhe, 1991), the former of which was comparable and the latter of which was higher compared to this study. However, statistical results from Leck and Rodhe (1991) indicated that no single factor such as salinity or certain phytoplankton species could be accounted for these higher concentrations of DMS in the Kattegat/Skagerrak. Leck and Rodhe (1991) suggested that increased eutrophication of coastal regions may result in a net positive effect on DMS production in the Baltic Sea. The study

from Orlikowska and Schulz-Bull (2009) in the Bay of Mecklenburg (southern Baltic Sea) showed DMS concentration in the range from $< 0.3$ nmol $L^{-1}$ in November 2008 up to 120 nmol $L^{-1}$ in May 2008. Considering that the concurrent Chl $a$ concentrations from phytoplankton were only 2–4 µg $L^{-1}$, Orlikowska and Schulz-Bull (2009) proposed that macroalgae could also contribute significantly to the DMS production. A comparison with data from other times-series coastal areas (Table 2) reveals that mixed layer DMS concentrations at BE are generally comparable to those measured at other time-series stations

in coastal regions like in the Baltic Sea, the Mediterranean Sea or the Indian Ocean. $DMSP_t$ concentrations from BE are in the same range as the concentrations reported from the NW Mediterranean Sea and the western English Channel, but they are lower than those reported from the southern North Sea (including the Belgian and Dutch coasts), the Revellata Bay (Gulf of Calvi, Mediterranean Sea) and the coast off Goa (eastern Arabian Sea, Indian Ocean). $DMSO_t$ concentrations at BE were generally in the same range as reported from other time series sites except for the extremely high $DMSO_t$ concentrations

measured at the coast of Belgium (southern North Sea). The obvious high variabilities in the range of DMSP(O) concentrations are most probably resulting from the interplay of various factors such as differences in sampling periods/frequency, the prevailing phytoplankton/bacteria community composition and succession, and the eutrophication status as well as the occurrence of anoxic events.

**Table 1. Spearman's rank correlation coefficients of correlations of all sulphur compounds with several ambient parameters, as well as algae groups in the mixed layer at the BE station during 2009–2018. Only datasets were used for which all environmental parameters (n = 85), phytoplankton data (n = 61 for Diat and Dino, n = 48 for Chryso and n = 35 for Prym) were available, and data were averaged for the mixed layer. Bold numbers indicate that a correlation is significant at the 0.01 level (two tailed). Diat, Dino, Prym and Chryso, stand for diatoms, dinoflagellates, prymnesiophyceae and chrysophytes, respectively. N:P ratios stand for the**
**ratio of the sum of nitrate, nitrite and ammonium to phosphate.**

|  | DMS | $DMSP_t$ | $DMSP_p$ | $DMSP_d$ | $DMSO_p$ | $DMSO_d$ | $DMSO_t$ |
|---|---|---|---|---|---|---|---|
| Chl $a$ | **-0.31** | 0.19 | 0.11 | 0.04 | **0.29** | -0.11 | 0.2 |
| Temperature | **0.41** | 0.11 | 0.14 | -0.12 | 0.26 | 0.12 | 0.24 |
| Salinity | -0.19 | 0.08 | -0.04 | -0.05 | -0.09 | -0.11 | -0.08 |
| N:P | -0.15 | -0.21 | -0.08 | -0.04 | -0.12 | -0.04 | 0.14 |
| Diat | -0.04 | -0.05 | -0.13 | 0.16 | -0.25 | -0.24 | -0.27 |
| Dino | 0.10 | 0.09 | 0.19 | -0.26 | 0.2 | 0.06 | 0.19 |
| Prym | 0.25 | **0.38** | **0.47** | 0.14 | **0.35** | 0.30 | **0.34** |
| Chryso | 0.32 | **0.44** | **0.37** | 0.29 | 0.28 | 0.25 | 0.32 |
| $DMSO_t$ | **0.35** | **0.79** | **0.72** | **0.43** | **0.86** | **0.75** |  |





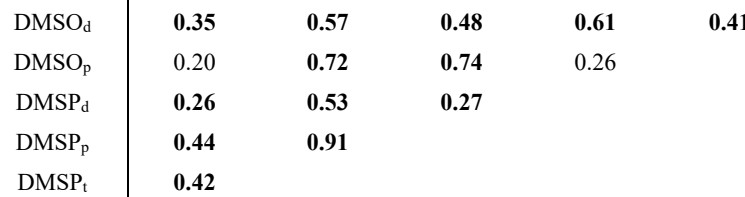

| | | | | | |
|---|---|---|---|---|---|
| DMSO$_d$ | **0.35** | **0.57** | **0.48** | **0.61** | **0.41** |
| DMSO$_p$ | 0.20 | **0.72** | **0.74** | 0.26 | |
| DMSP$_d$ | **0.26** | **0.53** | **0.27** | | |
| DMSP$_p$ | **0.44** | **0.91** | | | |
| DMSP$_t$ | **0.42** | | | | |

Figure 3: Monthly and mean seasonal distributions of chlorophyll a (a, b), DMSP$_p$ (c, d), DMSP$_d$ (e, f), DMS (g, h), DMSO$_p$ (i, j) and DMSO$_d$ (k, l) at the BE station during 2009–2018. Black dots and the black line (g and h) indicate monthly and mean seasonal distributions of the MLD, respectively. The black arrows (g) indicate the Major Baltic Inflow (MBI) events in November 2010 and December 2014, respectively. The red arrows indicate elevated concentrations of DMS under hypoxia/anoxia in 2009, 2010, 2016





and 2018, respectively. The concentrations shown in the left panels are given on a natural logarithmic scale. In 2009, DMSO data were only available from April to July.


**Table 2. Surface sulphur compounds concentrations from coastal time-series studies.**

| Region | Period of sampling | Sampling frequency | DMS Avg. Range (min–max) | DMSP$_t$ Avg. Range (min–max) | DMSO$_t$ Range (min–max) | Chl $a$ Avg. Range (min–max) | Reference |
|---|---|---|---|---|---|---|---|
| Boknis Eck Time-Series Station, the Southwest Baltic Sea | Apr 2009–Dec 2018 | monthly | 1.7[a] 0.1–12.2[a] | 18.5[a] 1.4–85.4[a] | 31.1[a] 2.5–209.8[a] | 2.1[a] 0.3–10.8[a] | This study |
| Station B1, Himmerfjärden Fjord, the West Baltic Sea | Jan 1987–Jun 1988 | weekly in spring, biweekly in summer and monthly in winter | 1.51 0.1–6.3 | nm | nm | ng < 1–12 | Leck et al. (1990) |
| Heiligendamm station, Bay of Mecklenburg, the Baltic Sea | Jan-Nov 2008 | weekly | ng up to ~ 120 | nm | nm | ng ~ 1–7 | Orlikowska and Schulz-Bull (2009) |
| The Southern North Sea | Feb–Oct 1989 | monthly | 3.92 0.1–> 50 | ng up to 450 | nm | ng up to 35 | Turner et al. (1996) |
| The Belgian Coastal Zone, the North Sea | Jan–Dec 2016 | bimonthly from Feb to Jun and monthly for the rest | ng up to 250 | ng up to 1740[b] | ng up to 620 | ng up to 36 | Speeckaert et al. (2018) |



| Location | Period | Frequency | | | | | Reference |
|---|---|---|---|---|---|---|---|
| Coast of Den Helder, The Netherlands | Nov 1991–Nov 1992 and Jan–Jun 1993 | Biweekly in 1991 and 1992, more frequent in 1993 | ng<br>0–18 | ng<br>7– > 1500[b] | nm | ng<br>0–65 | Kwint and Kramer (1996) |
| Station L4, the Western English Channel | May–Oct 2014 | weekly | 5.1<br>up to 17 | ng<br>~ 10–100 | ng<br>2.3–102 | ng<br>~ 0.1–2.4 | Dixon et al. (2020) |
| Toulon Bay, the NW Mediterranean Sea | Jan–Dec 1997 | monthly | 9.8<br>3.6–21.03 | nm | nm | ng<br>0.2–2.5 | Despiau et al. (2002) |
| The NW Mediterranean Sea | Jan 2003–June 2004 | monthly | ng<br>~ 0.5–19 | ng<br>~ 10–71.7[b] | ng<br>~ 0–24.2[b] | ng<br>~ 0.4–2.8 | Vila-Costa et al. (2008) |
| Revellata Bay, Gulf of Calvi | April 2015–Jul 2016 | weekly to biweekly | nm | 130[c]<br>62–205[c] | 4.9[c]<br>1.5–8.6[c] | ng | Richir et al. (2019) |
| The Zuari estuary off Goa | Dec 1999–Jan 2001 | monthly | 5.8<br>0.3–15.4 | 68.3<br>0.8–415.9 | nm | ng<br>up to ~ 10 | Shenoy and Patil (2003) |
| Candolim Time-Series station, coast off Goa | Sep 2009–Dec 2013 | monthly | 22.5<br>0.5–442 | 24<br>0.4–252 | 27.8<br>0.6–185.9 | ng<br>0.1–14.4 | Bepari et al. (2020) |
| Rothera Time-Series Station, Ryder Bay, West Antarctic | Sep 2012–Mar 2017 | 2–3 times/week | 3.7<br>0.1–170 | ng | ng | ng | Webb et al. (2019) |

[a] averaged for the mixed layer; [b] given as $DMSP_p$ or $DMSO_p$; ng and nm stand for not given and not measured, respectively. [c] given as $\mu mol.g_{fw}^{-1}$. The units of sulphur compounds and Chl a are given as nmol $L^{-1}$ and µg $L^{-1}$, respectively.





## 4.2 Temporal trend analysis

Temporal trend analysis was calculated by anomaly detection via subtracting the overall monthly mean (2009–2018) from the individual monthly mean, followed by smoothing with a 12-point moving average, which was used to reduce the effects of the seasonal as well as annual cycles on the temporal trend. Temperatures showed increasing trends in the mixed layer and the bottom layer (Fig. 4a) during our study. The trends were 0.2 °C yr$^{-1}$ and 0.1 °C yr$^{-1}$ (Table 3) in the mixed layer and the bottom layer, respectively. Our result is consistent with the study by Belkin (2009), who reported a post-1987 warming rate in the

Baltic Sea exceeding 1.0 °C decade$^{-1}$. A less pronounced trend of 0.02 °C yr$^{-1}$ (in 1 m and 25 m) during 1957–2013 was reported by Lennartz et al. (2014). This disagreement may arise from the fact that the result reported by Lennartz et al. (2014) covers a much longer study period and there might have an acceleration trend of increasing temperature started around 2014 (Rahmstorf et al., 2017). Salinity in the bottom layer (25 m; Fig. 4b) did not show significant trends in our study, which is in agreement with Lennartz et al. (2014). Additionally, there is no trend of dissolved oxygen in the bottom layer (Fig. 4b), which is different

to the trend computed by Lennartz et al. (2014) who reported bottom $O_2$ concentrations were decreasing over 56 years. Again, this difference is attributed to the much shorter observation period of our study compared to Lennartz et al. (2014). Dissolved inorganic nitrogen (DIN; Fig. 4c) and phosphate (Fig. 4d) showed slightly increasing trends of 0.5 µmol L$^{-1}$ yr$^{-1}$ and 0.2 µmol L$^{-1}$ yr$^{-1}$ (Table 3) in the bottom layer; but similar trends were not observed or apparent in the mixed layer. The decreasing trends for nutrients (in 1 m and 25 m) at BE reported by Lennartz et al. (2014) are due to a reduction of nutrient inputs to the

Baltic Sea to improve the eutrophication status of the Baltic Sea (HELCOM, 2018a). This is consistent with Kuss et al. (2020) who reported a decline in DIN from 1995–2004 and TP (total phosphate) from 2005–2009 in the Belt Sea with no significant changes thereafter. In our study, the increasing trends of nutrients in 25 m coincided with increasing temperature as well as more frequent hypoxic/anoxic events (i.e. ongoing deoxygenation) in 25 m. Increasing temperature favoured bacteria decomposing actives beneath the thermocline due to more pronounced water column stratification, supporting their

remineralisation and thus, leading more consumption of dissolved oxygen, and releasing more nutrients in the bottom layer (Hoppe et al., 2013; Lennartz et al., 2014). Thus, the increasing nutrient concentrations are not a general eutrophication of the water column but a natural effect limited within the bottom layer. Chl $a$ concentrations (Fig. 4e) showed an increasing trend of 0.2 µg L$^{-1}$ yr$^{-1}$ (Table 3) in the mixed layer, which are primarily driven by high concentrations in 2017, similar to the variability of the sum of fucoxanthin (a marker pigment for diatoms) and peridinin (a marker pigment for dinoflagellates; Fig.

4i).

    DMSP$_t$ concentrations (Fig. 4f) showed a slightly decreasing trend both in the mixed layer (-0.9 nmol L$^{-1}$ yr$^{-1}$) and in the bottom layer (-0.3 nmol L$^{-1}$ yr$^{-1}$; Table 3), as opposed to the upward trends of Chl $a$ in the mixed layer and temperature both in the mixed and bottom layer. A similar decreasing trend in the mixed layer (-9.2 ng L$^{-1}$ yr$^{-1}$) were detected for the sum of the pigments 19′-hexanoyloxyfucoxanthin (19'-hex, a marker pigment for prymnesiophyceae) and 19′-butanoyloxy fucoxanthin

(19'-but, a marker pigment for chrysophytes; Fig. 4i). This indicates that the general trend of DMSP$_t$ concentrations in the mixed layer might be primarily controlled by the productivity of chrysophytes and prymnesiophyceae (see also Table 1). The





decreasing trend for DMS ($-0.1$ nmol $L^{-1}$ $yr^{-1}$; Fig. 4g) generally followed the pattern of $DMSP_t$ in the mixed layer and indicates that DMSP cleavage might play a dominant role in the production of DMS (see Table 1). Although no significant trend was observed for $DMSO_t$ (Fig. 4h), its general variability over time was similar to those of $DMSP_t$ and DMS in the mixed layer.

The decreasing trend observed for $DMSP_t$ in 25 m might be mainly attributed to the corresponding sinking particles from the mixed layer, as no trends were observed for Chl *a* or other algae groups in 25 m.

As a statistical test to decipher significant monotonic long-term trends in time series, the Mann-Kendall test (MKT) was also applied to detect the temporal trends of the individual months. However, no significant trends were observed for any of the sulphur compounds by the MKT test in our study.





**Figure 4: Temporal trends of anomalies of temperature (°C), dissolved oxygen (µmol L$^{-1}$), salinity, dissolved inorganic nitrogen (DIN; µmol L$^{-1}$), phosphate (µmol L$^{-1}$), Chl $a$ (µg L$^{-1}$), DMSP$_t$ (nmol L$^{-1}$), DMS (nmol L$^{-1}$), DMSO$_t$ (nmol L$^{-1}$), the sum pigments concentrations of Fuco and Peri (ng L$^{-1}$), 19′-hex and 19′-but (ng L$^{-1}$). Fuco stands for fucoxanthin, Peri stands for peridinin, 19′-hex stands for 19′-hexanoyloxyfucoxanthin and 19′-but stands for 19′-butanoyloxy fucoxanthin. The shaded areas indicate 95 %**





confidence intervals. Note that gaps were filled by linear interpolation in the case of one or two missing months in a row and large
       gaps between August to December 2009 in DMSO$_t$ (h) were filled by replacement with the median of the corresponding month.

**Table 3.** Statistics of the linear regression of the temporal trends for the anomalies of temperature (°C), dissolved oxygen (μmol L⁻¹), salinity, dissolved inorganic nitrogen (DIN; μmol L⁻¹), phosphate (μmol L⁻¹), Chl $a$ (μg L⁻¹), DMSP$_t$ (nmol L⁻¹), DMS (nmol L⁻¹), DMSO$_t$ (nmol L⁻¹), the sum of pigments of Fuco and Peri ((ng L⁻¹), Hex and But ((ng L⁻¹). $r^2$: coefficient of determination in the simple linear regression calculated by the time-series monthly individual parameters. Sen's slope: median slope present in time series (yr⁻¹) according to Sen (1968).

| | Mixed layer | | | | Bottom layer (25 m) | | | |
|---|---|---|---|---|---|---|---|---|
| | $r^2$ | $p$ value | Sen's slope | n | $r^2$ | $p$ value | Sen's slope | n |
| Temperature | 0.54 | < 0.01 | 0.2 | 108 | 0.35 | < 0.01 | 0.1 | 108 |
| Oxygen | 0.14 | < 0.01 | -1.3 | 108 | < 0.01 | 0.67 | -1.2 | 108 |
| Sal | < 0.01 | 0.58 | 0.1 | 108 | 0.03 | 0.07 | 0 | 108 |
| DIN | < 0.01 | 0.89 | 0 | 108 | 0.21 | < 0.01 | 0.5 | 108 |
| Phosphate | 0.27 | <0.01 | 0 | 108 | 0.22 | < 0.01 | 0.2 | 108 |
| Chl $a$ | 0.40 | < 0.01 | 0.2 | 105 | < 0.01 | 0.53 | 0 | 105 |
| DMSP$_t$ | 0.17 | < 0.01 | -0.9 | 107 | 0.21 | < 0.01 | -0.3 | 107 |
| DMS | 0.12 | 0.01 | -0.1 | 107 | 0.17 | < 0.01 | 0 | 107 |
| DMSO$_t$ | 0.01 | 0.23 | 0.3 | 105 | < 0.01 | 0.45 | -0.2 | 105 |
| Fuco+Peri | 0.02 | 0.18 | -33.3 | 105 | 0.07 | 0.06 | -30.2 | *105* |
| 19´-Hex+19´-But | 0.23 | < 0.01 | -9.2 | 105 | *NA* | *NA* | *NA* | *43* |

## 4.3 Influence of extreme events at BE on the sulphur compounds

### 4.3.1 The Major Baltic Inflow events

A MBI event lasted for one month in 2014 and was detected in the Eckernförde Bay by elevated sea levels after an outflow
       period, which indicated that its inflow began on 10 December 2014 (Ma et al., 2020). Therefore, the sampling at BE on 16
       December 2014, took place during the MBI period. Our results show that the sulphur compounds concentrations in the water
       column in December 2014 and in January 2015 were low and similar to the overall mean concentrations of sulphur compounds
       in December/January for the period 2009-2018, indicating that the MBI in December 2014 did not influence the concentrations
of sulphur compounds at BE directly. Relatively higher DIN and dissolved phosphate concentrations (Fig. 2g and 2i) were





measured in December 2014, and this would be assumed to trigger a more significant spring bloom in the next year and therefore, higher sulphur compounds concentrations. Indeed we measured higher concentrations of sulphur compounds in March and April 2015; however, this is probably attributed to the unusually higher proportion of prymnesiophyceae of the phytoplankton community (see. Sect. 4.4), and this high fraction of prymnesiophyceae was not supposed to be caused by the

rich nutrients accumulated in December 2014. The peak of the spring bloom in 2015 could not be identified considering moderate Chl $a$ concentrations in February (1.0 µg L⁻¹) and March (2.0 µg L⁻¹), but a substantial decrease of nutrients occurred between February and March 2015. Concentrations of DIN and dissolved phosphate stayed high until February 2015. Subsequently, DIN concentrations decreased from 8.0 µmol L⁻¹ in the mixed layer on 23 February to 0.1 µmol L⁻¹ on 17 March, with dissolved phosphate from 0.7 to 0.1 µmol L⁻¹ in the same case. Depleted nutrients in March suggested the spring bloom

peak between the sampling date in February and in March 2015 was apparently not captured by our monthly measurements and underlines the necessity of high frequent sampling. As a minor algae group at BE, prymnesiophyceae tends to occur to accumulate towards the end of spring diatom blooms in oligotrophic conditions (Veldhuis et al., 1986), and this was confirmed by the decreasing concentration of silicate from 12.5 µmol L⁻¹ in the mixed layer in February 2015 to 2.2 µmol L⁻¹ in March, which is the limiting growth factor of diatoms. Therefore, we conclude that the accumulation of nutrients had been consumed

by diatoms between February and March before prymnesiophyceae formed the bloom. However, the much higher than usual relative abundance of prymnesiophyceae in March and April 2015 (see Figure. 5a) might have been transported to BE by the saline water from the North Sea, where prymnesiophyceae are abundant (Speeckaert et al., 2018).

Another relatively weak MBI occurred in late autumn 2010 (Mohrholz et al., 2015), and we measured elevated salinity concentrations in November 2010 at BE (see Fig. 2c). Subsequently, concentrations above-average for DMS (1.9–3.7 nmol L⁻¹),

DMSP$_p$ (50.9–84.5 nmol L⁻¹) and DMSO$_p$ (32.2–40.6 nmol L⁻¹) were measured in spring bloom during February – April 2011, coincided with the exceptionally higher relative abundance of chrysophytes in the mixed layer (see Fig. 4). We assume that this chrysophyte was rather new and uncommon only occurred in Kiel Bight and Mecklenburg (Wasmund et al., 2012). Therefore, it is possible that this uncommon chrysophyte was brought into the western Baltic Sea via saline waters in autumn 2010 and bloomed in spring 2011, resulting in high concentrations of DMSP and thus DMS(O) at BE.

Overall, enhanced DMSP$_p$ concentrations (> 50 nmol L⁻¹) measured during the spring bloom in 2011 and 2015, both followed after the MBI events in winter and comprised new-forming phytoplankton groups not common at BE. Therefore, we hypothesize that MBI was likely to influence sulphur compounds concentrations by introducing new phytoplankton species which are good DMSP producers.

### 4.3.2 Low oxygen events

During seasonal hypoxic/anoxic events at BE (see Fig. 3e), elevated concentrations of DMS (up to 4.19 nmol L⁻¹) were measured in the bottom layer in August 2009, August–October 2010, September 2016 and September 2018 (see Fig. 3g). These elevated DMS concentrations (2.3 ± 1.4 nmol L⁻¹) in the bottom layer (20–25 m) were generally comparable to or lower than those found in the mixed layer (0-5 m; 3.4 ± 2.2 nmol L⁻¹), but higher than those in the overlying water layers(15–20 m;





1.2 ± 1.2 nmol L$^{-1}$). Shenoy et al. (2012) reported extremely high concentrations of DMS (up to 442 nmol L$^{-1}$) as well as
enhanced DMSP$_t$, DMSO$_t$ and methanethiol concentrations in the bottom layer during an anoxic event at Candolim Time-
Series Station (CaTS) off Goa, West India, in September 2009 and suggested that this unusually high DMS concentration
might result from a combination of sources such as DMSP cleavage, DMSO reduction, methylation of methanethiol and
hydrogen sulphide under anoxic conditions. Later on, Bepari et al. (2020) observed high concentrations of DMS (233 nmol L$^{-1}$)
in the bottom layer during an anoxic event at CaTS in September 2013 and assumed that sediments might also be an important
source of DMS, additional to the breakdown of simultaneously high concentrations of DMSP$_t$ (206–252 nmol L$^{-1}$) in the water
column. However, in the case of BE, concentrations of DMSP$_t$ (4.7 ± 4.9 nmol L$^{-1}$) or DMSO$_t$ (4.1 ± 2.2 nmol L$^{-1}$) measured
in the bottom layer during hypoxia/anoxia events were lower to those in the mixed layer (DMSP$_t$: 20.6 ± 8.1nmol L$^{-1}$and
DMSO$_t$: 32.4 ± 17.0 nmol L$^{-1}$) or the overlying water layer (DMSP$_t$: 11.9 ± 4.1nmol L$^{-1}$ and DMSO$_t$: 17.7 ± 11.6 nmol L$^{-1}$),
which indicate that DMSP cleavage or DMSO reduction processes seem unlikely to account for the main fraction of DMS
production. Therefore, it is reasonable to assume that these elevated concentrations of DMS in the bottom layer might have
been at least in part released from the sediments (Nedwell et al., 1994) and might originate from the methylation of
methanethiol (Song et al., 2020). However, elevated DMS concentrations in the bottom layer were not always measured
simultaneously during low oxygen events. In only 5 out of 18 sampling months, we observed elevated DMS concentrations
together during low oxygen events (see Fig. 3g and 3e). Therefore, we speculate that there is a switch between DMS generation
and removal processes in the sediments (Kiene, 1988; Nedwell et al., 1994), which needs to be further investigated at BE.

**4.4 Relationships between the sulphur compounds and phytoplankton groups**

In general, phytoplankton composition and succession (Fig. 5a) at the BE station were similar to previous studies from the
Baltic Sea with a recurrent pattern of diatoms dominating the bloom in spring (February–April) and summer (June–August)
followed by dinoflagellates in autumn (September–November) (Smetacek, 1985; Wasmund et al., 2008). Diatoms were the
most dominant phytoplankton group at the Boknis Eck station, especially during the spring bloom and reached their maximum
in March. The fraction of diatoms gradually decreased in April and May whereas the fractions of prymnesiophyceae,
cryptophytes and chlorophytes increased, accompanied by the development of cyanobacteria. Minor summer blooms most
commonly occurred in August below the surface water (e.g. in 15 m or 20 m) at BE, as a result of stratification which restricted
the bottom nutrients supply to the surface layer (Fig. 5b). The autumn/winter bloom period (September–December) was mainly
composed of a mixture of dinoflagellates and diatoms or as a succession of these two groups. Overall, diatoms and
dinoflagellates were the most common phytoplankton groups at BE.



### 4.4.1 Relationship between sulphur compounds and phytoplankton groups

Positive correlations were found between chrysophytes and $DMSP_p$ as well as prymnesiophyceae and $DMSP_p$ in the mixed layer (Table 1). Enhanced concentrations of $DMSP_p$ (> 50 nmol $L^{-1}$) were associated with the high relative abundance of

chrysophytes (25 %–62 % between February and April 2011) and prymnesiophyceae (29 %–56 % in March and April 2015, respectively) in the mixed layer. Reports from Wasmund et al. (2012) and Wasmund et al. (2016) confirmed that these two

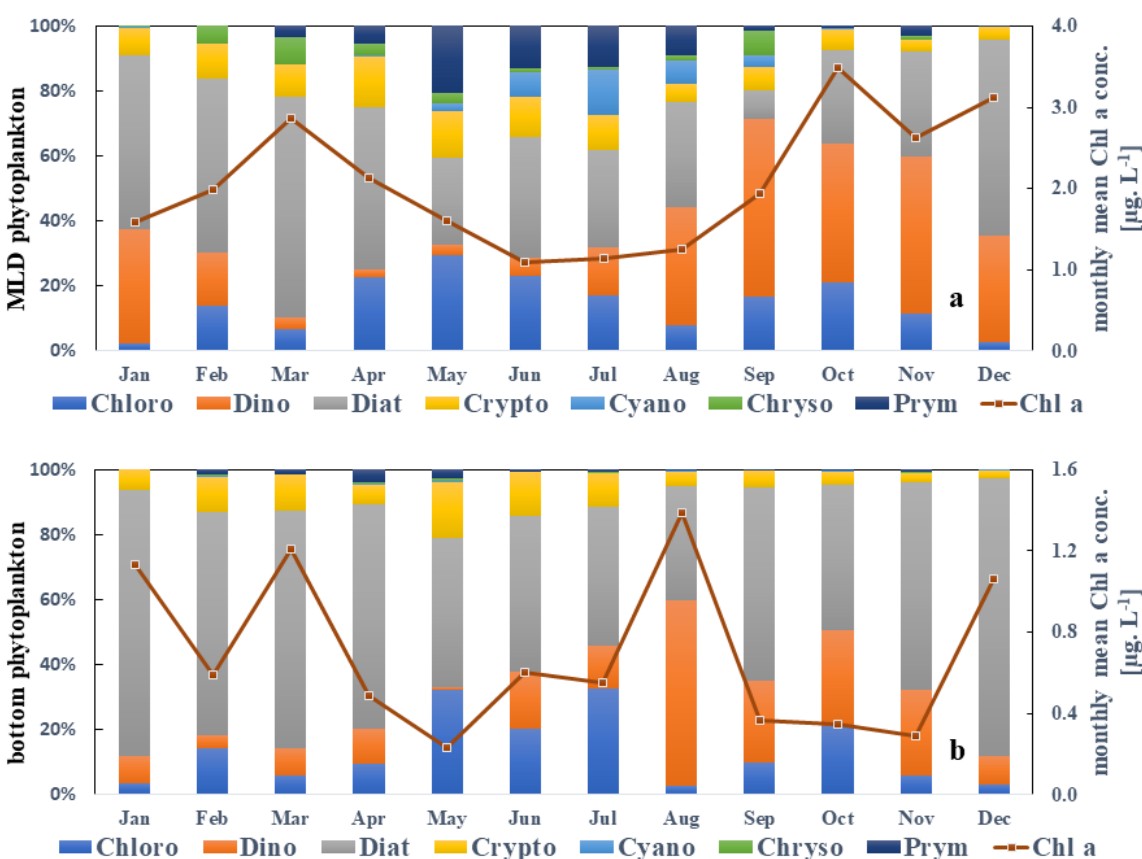

**Figure 5: Mean seasonal phytoplankton composition at the Boknis Eck station from 2009 to 2018 based on the result of Chemtax. The same months of each year were averaged. Please note that pigments samples are available from April 2009 until December 2018.**

**The values are averaged for the mixed layer (a) and bottom layer (b), respectively. Prym: prymnesiophyceae, Chryso: chrysophytes, Cyano: cyanobacteria, Crypto stands for cryptophytes, Diat stands for diatoms, Dino stands for dinoflagellates and Chloro stands for chlorophytes. The red line indicates monthly mean Chl *a* concentrations for the mixed layer and bottom layer, respectively.**

algae groups were higher in their abundances in the years 2011 and 2015 in the western Baltic Sea, respectively. Our results suggest that these two algae groups might be the main producers of DMSP in the mixed layer at BE and this is in agreement

with the results of the previous studies of Keller et al. (1989) and Belviso et al. (2001) who found that chrysophytes and prymnesiophyceae can be significant DMSP producers in general. No correlation was found between dinoflagellates and $DMSP_p$ in the mixed layer (Table 1). In previous studies, massive dinoflagellates blooms were reported to be closely coupled





with high concentrations of DMSP. For example, the highest DMSP concentration (4240 nmol $L^{-1}$) reported so far was found tightly linked to elevated anduance of *A. sanguinea* (Kiene et al., 2019). This could be attributed to that the ability to produce

DMSP is considerably variable among different genus and species (Keller et al., 1989). Hence, low or high DMSP concentrations during dinoflagellate blooms are dependent on the dominant species or composition. Typically, *Ceratium* spp was one of the most common genera during dinoflagellate-dominant autumn blooms in the western Baltic (Wasmund et al., 2015). However, the ability of *Ceratium* spp to produce DMSP is rather weak compared to other species or genus belonging to dinoflagellates (Keller et al., 2012). The discrepancy between maximum Chl *a* concentration (12.4 μg $L^{-1}$) dominated by

dinoflagellates and the $DMSP_p$ concentrations (25.2 nmol $L^{-1}$) in 1 m depth in October 2017, might be attributed to that *Ceratium tripos* was the dominative species during dinoflagellates bloom (Wasmund et al., 2018), which might be of minor importance for the DMSP pool at BE. Positive correlations were found between prymnesiophyceae and $DMSO_p$ (Table 1) in the mixed layer. Similar to $DMSP_p$, enhanced concentrations of $DMSO_p$ (> 80 nmol $L^{-1}$) were measured with high proportions of prymnesiophyceae in March and April 2015, suggesting prymnesiophyceae might also be important producers of DMSO at

BE. Overall, despite prymnesiophyceae and or chrysophytes were good producers of DMSP(O) at BE, the seasonal distributions of $DMSP(O)_p$ in the mixed layer followed that of Chl *a* instead of specific algae groups in terms of their large interannual/seasonal variabilities (Fig. 6a).

$DMSP_p$ and $DMSO_p$ concentrations in the bottom layer (25 m) were generally low throughout the year except for August. We observed a higher relative abundance of dinoflagellates in 25 m in August (Fig. 6b) who were probably more adapted to

seawater stratification (Estrada et al., 1985). The ability of vertical migration of dinoflagellates helps them to cross the pycnocline to get access to the nutrients which accumulate below the mixed layer during the periods of the pronounced summer stratification. Better nutrient access can promote the metabolic activity and thus the DMSP production within dinoflagellates. Also, as mentioned above, the ability to produce DMSP among dinoflagellates varies substantially. For instance, the elevated concentrations of $DMSP_p$ in 25 m in August 2011, 2012 and 2014 might result from the observed high biomass of *Alexandrium*

spp in the phytoplankton community in the Kiel bay (Wasmund et al., 2012; Wasmund et al., 2013, 2015), which is generally considered as good DMSP producer in dinoflagellates (Caruana and Malin, 2014). Therefore, the relationship between dinoflagellates and DMSP at BE may not be well-represented at the class levels (Griffiths et al., 2020).

DMS concentrations were negatively correlated with Chl *a* concentrations and poorly correlated with any phytoplankton groups (Table 1) in the mixed layer at BE. Similar cases for these correlations have been reported in many studies (Townsend

and Keller, 1996; Toole and Siegel, 2004) due to the complex production and removal processes of DMS.





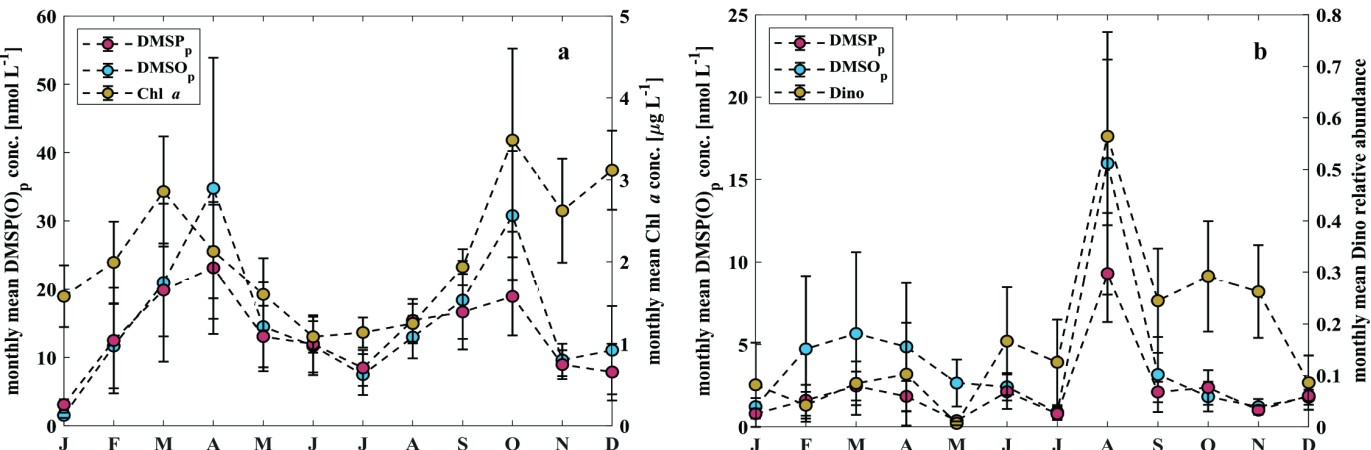

**Figure 6: (a): monthly mean DMSP$_p$, DMSO$_p$ and Chl *a* concentrations in the mixed layer during 2009–2018 at BE. (b): monthly mean DMSP$_p$, DMSO$_p$ concentrations and relative abundance of dinoflagellates in 25 m during 2009–2018 at BE. Error bars represent standard errors of the mean of the samples.**

### 4.4.2 Predictive algorithms

An algorithm which is able to predict DMS concentrations and thus its emission to the atmosphere could potentially help to
improve climate models (Simó and Dachs, 2002; Wang et al., 2020). To reproduce and predict DMS(P) concentrations, parameters such as Chl *a*, temperature, solar radiation or nutrients are often used. To this end, we tested three predictive algorithms suggested by Simó and Dachs (2002) (S02) and Watanabe et al. (2007) (W07) as well as Nagao et al. (2018) (N18) to predict DMS concentrations and the DMSP$_p$:Chl *a* ratios in the surface layer (5 m) at BE, respectively. The algorithm proposed by Simó and Dachs (2002) makes use of the mixed layer depth (MLD) and the MLD:Chl *a* ratio to predict DMS
concentrations in the mixed layer. Watanabe et al. (2007) proposed an empirical equation for the prediction of sea surface DMS concentrations by combining sea surface temperatures, nitrate and latitude. No significant correlations were found between the measured DMS concentrations from this study and the predicted DMS concentrations by applying the S02 and W07 algorithms. Possible reasons might be due to that S02 was derived from a global dataset from coastal and open ocean regions, and W07 was based on a dataset from the North Pacific open Ocean, which is in contrast to our coastal dataset.

The Fp ratio was first proposed by Claustre (1994) and was defined as a trophic status ratio. Then, inspired by Aumont et al. (2002), Nagao et al. (2018) proposed new Fp ratios representing the fractions of high and low DMSP producers in the phytoplankton community to predict the DMSP$_p$:Chl *a* ratios by using phytoplankton pigments:

Fp (high) = (19′-hex + 19′-but + peridinin) / ∑ pigments, (1)

Fp (low) = (fucoxanthin + zeaxanthin + alloxanthin + Chl b) / ∑ pigments, (2)

where ∑ pigments stands for the sum of fucoxanthin, peridinin, 19′-hex, 19′-but, zeaxanthin, alloxanthin and Chl b.

However, results from equations (1) and (2) did not work well with the DMSP$_p$:Chl a ratios from our study (Fig. 7a), neither the Fp (high) or Fp (low) ratios correlated well with the DMSP$_p$:Chl *a* ratios in 5 m. As discussed above, the ability to produce





DMSP for dinoflagellates was generally low at BE in the mixed layer. Therefore, we modified equations (2) and (3) by moving peridinin from Fp (high) to Fp (low) as follows:

New Fp (low) = (fucoxanthin + zeaxanthin + alloxanthin + Chl b + peridinin) / $\sum$ pigments,    (3)

New Fp (high) $=$ $(19' - \text{hex} + 19' - \text{but})$ / $\sum$ pigments,    (4)

Significantly negative and positive correlations were found between the DMSP$_p$:Chl $a$ ratios and the new Fp (high) and new Fp (low) ratios in 5 m, respectively (Fig. 7b). The newly defined Fp (high) and Fp (low) represent the measured DMSP$_p$:Chl $a$ ratios accurately, additionally showing that DMSP$_p$ is mainly driven by the phytoplankton community. Then annual mean


Figure 7: (a): DMSP$_p$:Chl $a$ vs. Fp ratio (from Nagao et al. (2018)) in 5 m for the period 2009-2018. (b): DMSP$_p$:Chl $a$ vs. modified Fp ratio (new Fp ratio) in 5 m for the period 2009-2018: red dots depict Fp (high) (y = 72.90x + 3.78, $R^2$ = 0.54, $p < 0.01$, n = 65) and green dots depict Fp (low) (y = -48.33x + 51.35, $R^2$ = 0.46, $p < 0.01$, n = 65). (c): annual mean measured DMSP$_p$ vs. annual mean predicted DMSP$_p$ concentrations in 5 m for each year during 2009–2018: y = 0.82x + 4.78, $R^2$ = 0.86, $p < 0.01$, n = 9. Note that the
red dot was not included into the linear regression line (blue) and the dash line indicates the identity (1:1) line.



DMSP$_p$ concentrations in 5 m were simulated by annual mean 19′-hex, 19′-but and Chl a concentrations, and were compared to the measured concentrations (Fig. 7c). Our simulated DMSP$_p$ is in good agreement with our measured DMSP$_P$ except for the year 2017 (the red dot in Fig. 7c). In 2017, we measured the most pronounced spring (Chl $a$: 9.0 µg L$^{-1}$) and autumn blooms (Chl $a$: 12.4 µg L$^{-1}$) of the entire observation period. The blooms were dominated by diatoms and dinoflagellates,
which led to maximum annual mean Chl $a$ but low DMSP$_p$ concentrations.

## 5. Conclusions

We present a unique and comprehensive time-series study of sulphur compounds (DMS, DMSP and DMSO) at the Boknis Eck Times-Series Station, located in the Eckernförde Bay (SW Baltic Sea), from 2009 to 2018. Distinct interannual and seasonal variabilities of sulphur compounds were tightly linked to the phytoplankton composition at BE. DMSP$_p$ and DMSO$_p$
concentrations were generally enhanced in spring and autumn in the mixed layer, following the pattern of Chl $a$. Mixed layer DMSP$_t$ and DMS did not follow the increasing trends of the mixed layer temperature and Chl $a$ during the ten-year observation period. The main DMSP and DMS producers, namely, prymnesiophyceae and chrysophytes (represented by their marker pigments 19′-hex and 19′-but, respectively) decreased in their total abundances over the ten years.

    MBI events, which occurred in November 2010 and December 2014 at BE, might have influenced sulphur compounds
concentrations by introducing uncommon but important DMSP producers. Enhanced DMS concentrations in the bottom layer were measured during seasonal hypoxic/anoxic events, suggesting that sediment might be an important source of DMS for the overlying seawater. In contrast to the mixed layer, elevated concentrations of DMSP$_p$ and DMSO$_p$ usually occurred in the bottom layer in August at BE are due to specific dinoflagellate occurrence and stratification of the water column. Migrating dinoflagellates increased in their abundances due to nutrient-rich conditions in the deep layer and elevated light conditions in
the surface layer at BE. A modified algorithm, based on the phytoplankton pigments, shows an improvement to predict surface (5m) annual mean DMSP$_p$ concentrations at BE when compared with the original approach proposed by Nagao et al. (2018), highlighting the main drivers of DMSP dynamics at BE.

    Overall, the variabilities of sulphur compounds at BE were closely linked to a complex interplay of biotic and abiotic factors at BE. Continuous observations at BE, with an emphasis on algae and bacteria group identification together with their activities
determination, is of great importance (1) to capture the dynamics of DMS(P/O) and plankton community interactions and (2) to decipher the production pathways for sulphur compounds in the future, especially in view of the ongoing environmental changes such as ocean warming and acidification. Sediment samples from BE are also suggested to be collected in the future, as they are likely to contain high concentrations of sulphur compounds as previously reported (Williams et al., 2019). Moreover, an increasing frequency in sampling during seasonal phytoplankton blooms and low oxygen events will help to
capture the dynamic of sulphur compounds. The decadal observation at the BE time-series station shows how important long-term observations are to understand the local impacts and changes due to global warming and climate changes. We recommend



establishing more time-series stations and keeping existing stations running to observe and understand the impact of global changes worldwide on marine ecosystems.

*Data availability*. Data are available from the Boknis Eck database at https://www.bokniseck.de//database-access  (Bange and Malien, 2020).

*Author contributions*. YNZ, CS, DB and HWB designed the study and participated in the fieldwork. Sulphur compounds measurements and data processing were done by YNZ, CS and DB. YNZ conducted further data analysis and wrote the article with contributions from all co-authors.


*Competing interests*. The authors declare that they have no conflict of interest.

**Acknowledgments**

We thank the captain and crew of the RV *Littorina* and *Polarfuchs* as well as many colleagues and students (namely S. Marcks and J.P.M. Heyda) involved in the sampling and measurements of the Boknis Eck Time-Series Station. We especially thank
Kerstin Nachtigall for the phytoplankton pigments measurements and Frank Malien for the nutrients/dissolved oxygen measurements. The time-series station at BE was supported by DWK Meeresforschung (1957–1975), HELCOM (1979–1995), BMBF (1995–1999), the Institut für Meereskunde (1999–2003), IfM-GEOMAR (2004–2011) and GEOMAR (2012–present). The Boknis Eck Time-Series Station (www.bokniseck.de) is run by the Chemical Oceanography Research Unit of GEOMAR, Helmholtz Centre for Ocean Research Kiel. Data from Boknis Eck are available from www.bokniseck.de/database-access.
Yanan Zhao is grateful to the China Scholarship Council (CSC) for providing financial support (File No. 201606330066).

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
