# Peer review of "A decade of DMS, DMSP and DMSO measurements in the southwestern Baltic Sea"

_Biogeosciences, 2020_

## Referee Comment (RC1) · Anonymous Referee #1 · 2 Feb 2021

The manuscript by Zhao et al. deals with a long-term study (10 years data set) of the concentrations of selected dissolved and particulate sulfur species in the water column at the well-established Boknis Eck long-term observation station, Eckernförde Bight, at the transition between the North and the Baltic Sea. It addresses a so far under-investigated topic, clearly of interest for the international scientific BG community, of general biogeochemical relevance as well as consequences for the role of coastal ecosystems for the (trans)formation of climate-relevant gases. The manuscript is well written in most aspects, well structured, and I really enjoyed reading it and am waiting to see it published in a quality-improved final version. The manuscript presents original analytical data of high quality, links these results to phytoplankton indicators as well as hydrography dynamics to deepen our understanding for the controlling factors in this

part of the sulfur system. The scientific concept strongly relies on the concept of long-term data observations, but with a clear extension beyond a simple monitoring concept. Results are significant, well described, and discussed in terms of the use of literature-based knowledge. What I am missing here, however, is a reference to previous work published on benthic methane and sulfur cycling and the corresponding element fluxes from Eckernförde Bay, as well as a reference towards the potential impact of submarine ground water discharge. The authors are honest in their interpretation and the conclusions derived are justified by the sum of own and cited data. Title and abstract should be modified in a way to describe the actually measured components: For instance, no sulfide or sulfate data were measured or presented, but besides dissolved DMS compounds, also particulate phases were measured, which should be reflected by the title. Detailed comments: - L18: replace 'essential' by 'important' sources - L21: SCOPE studies on the topic? - L60: replace www links by reliable permanent sources - L123: approach (Ref?) - L127: 'hold' instead of 'true' - L495: Is there a better reference available? - Fig.1: Mention source for map details - Fig.2: Homogenize number format (e.g.'62.5') and bars given in parts e and f (captions) Mention how plots were generated (software) - Fig.6: What do you think about a further figure in comparing your results with previous studies? I would love to see that, indeed. - Fig.7: Replace all symbols by open ones, to allow for a recognition of the number of data generated - Fig.7 c: Start all text at the axes with capitals - Check the format of references: e.g. Wolfe et al. - Check for missing journals titles in the reference list: e.g., Toole et al., Rahmstorf et al., Richir et al., Song et al. . .

---

## Referee Comment (RC2) · Anonymous Referee #2 · 4 Feb 2021

General comments:

In the current study, the authors present a 10-year time series of various dissolved and particulate sulphur compounds collected in the Baltic Sea at the very well established long term monitoring station Boknis Eck. The data are presented and discussed in the context of various environmental data and phytoplankton groups, which were derived from marker pigments using the CHEMTAX$^®$ approach. The marker pigment approach is further explored in using an updated predictive equation to tackle future evolvements of the DMS, which would be beneficiary for modelling approaches. In addition, the time series was explored in the context of typical features observed in the Baltic Sea, anoxia and occasional ventilation events of the deep water based on salt-water advection from the North Sea. In this paper, the authors present a unique and high quality new data

set, which is valuable for a large community both interested in the sulphur compound cycle but also in the development of coastal environments, which are more and more under the pressure of climate change. Long-term data sets like this are essential to project climate change in coastal regions and beyond. The paper is very well structured and written. It was exciting to read and easy to follow. It gives an excellent overview of the available literature in the field. The authors have very carefully exploited their data and provided the appropriate statistical evaluation to support their results, and extracted the main findings of this study. So overall, the paper is in the range of very good to excellent and below you find a few suggestions, which should be suggest before final acceptance of the manuscript.

Specific comments

Line 12 and entire document: What is the rationale in using the class name for prymnesiophyceae, while all other groups were kept unspecific? Please harmonies all phytoplankton groups throughout the manuscript to prymnesiophytes.

Line 101 how long after sampling was the filtration done?

Line 116: Please give more details about the CHEMTAX$^{®}$ approach, did you use different depths, years etc. Please give input and output ratios for the data set.

Line 283 ff and Line 319ff Discussion about 4.3.1 The Major Baltic Inflow events and 4.3.2 Low oxygen events: The paper would benefit, if the two events would be better introduced at the beginning of each chapter including previous literature about associated findings with these type of events. This would be helpful to understand the rationale behind the expectation of changes also for the sulphur compounds.

Please check the references for consistency

---

## Author Comment (AC1) · 14 Feb 2021

We sincerely thank the referee for the valuable feedback that we used to improve the quality of this manuscript. According to these suggestions, we have supplemented several references and corrected some mistakes in our previous draft marked in yellow. Our point-by-point responses are listed below:

R1: ...What I am missing here, however, is a reference to previous work published on benthic methane and sulfur cycling and the corresponding element fluxes from Eckernförde Bay, as well as a reference towards the potential impact of submarine ground water discharge.

Authors: We sincerely appreciate the valuable comments. We have checked the literature carefully and added references (Bussmann et al., 1999; Bertics et al., 2013) on "sulphate reduction" and "submarine groundwater discharge" to section "4.3.2 Low oxygen events" of the ms, which would favour us to better explore various possibilities affecting bottom DMS concentrations at BE.

R2: . . . no sulfide or sulfate data were measured or presented, but besides dissolved DMS compounds, also particulate phases were measured, which should be reflected by the title.

Authors: Thank you for pointing this out. We have modified the title to "A decade of DMS, DMSP and DMSO measurements in the southwestern Baltic Sea", which now refers to both dissolved and particulate sulphur compounds. Corresponding changes have also been made in the abstract.

R3: L18: replace 'essential' by 'important' sources.

Authors: Done.

R4: L60: replace (www links) by reliable permanent sources.

Authors: www link was replaced with Lennartz et al. (2014).

R5: L127: 'hold' instead of 'true'.

Authors: Done.

R6: L21: SCOPE studies on the topic?

Authors: L21: Thank you for this point. Indeed, the SCOPE (Scientific Committee on Problems of the Environment) book series covers many aspects of the global sulfur cycle, see e.g., the studies presented in SCOPE books #19, #39, and #48. However, we think that the cited (more actual) references are better suited to justify our line of argument.

R7: L123: approach (Ref )?

Authors: We have added the reference of Reissmann et al. (2009).

R8: L495: Is there a better reference available?

Authors: The given link refers to the Boknis Eck database where all data discussed in our ms are archived.

R9: Fig.1: Mention source for map details.

Authors: We have added this information in the caption of Fig. 1.

R10: Fig.2: Homogenize number format (e.g.,'62.5') and bars given in parts e and f (captions) Mention how plots were generated (software).

Authors: According to the suggestion, we have re-adjusted number formats and bars in fig. 2e and 2f. In addition, we added the information on how the plots were generated in the captions of fig. 2 and fig. 3.

R11: Fig.6: What do you think about a further figure in comparing your results with previous studies? I would love to see that, indeed.

Authors: Thank you for your suggestion. In fig. 6 we provided mean seasonal variabilities of DMSPp, DMSOp and Chl a/relative dinoflagellate abundance in the mixed and bottom layers at BE, respectively. To our knowledge, this is the first time linking DMSPp and DMSOp variabilities to dinoflagellate abundance in the bottom layer, as most studies are focused on the surface/mixed layer (e.g., Table 2 in the ms). Among these time-series studies, only those provided simultaneous DMSP(O) and Chl a data could be compared to ours. Therefore, Vila-Costa et al. (2008), Speeckaert et al. (2018), Dixon et al. (2020) and Bepari et al. (2020) are included. Considering the unavailability of original data and/or different sampling strategies in the cited time series studies, we think these studies are not comparable to our study. Instead, we chose to make a comprehensive table (Table 2) which includes all existing time-series DMS/P/O data.

R12: Fig.7: Replace all symbols by open ones, to allow for a recognition of the number

of data generated - Fig.7 c: Start all text at the axes with capitals.

Authors: Yes, we have made all suggested modifications in the fig. 7 in the revised ms.

R13: Check the format of references: e.g. Wolfe et al. - Check for missing journals titles in the reference list: e.g., Toole et al., Rahmstorf et al., Richir et al., Song et al. . .

Authors: We feel sorry for our carelessness. We have corrected these references and checked others as well.
* * *
[Figure]

**Fig. 1.** Figure. 1...The location map was created with the m_map package for Matlab R2019 (Pawlowicz, 2020).

[Figure]

**Fig. 2.** Figure. 2...Time-depth Hovmöller Diagrams were generated with Matlab...

[Figure]

**Fig. 3.** Figure. 7...(c): Annual mean measured DMSPp vs. annual mean predicted DMSPp concentrations in 5 m for each year during 2009–2018.

---

## Author Comment (AC2) · 14 Feb 2021

We feel great thanks for the referee working on our manuscript and these comments are really helpful to improve the quality of this ms. According to these nice and precise suggestions, we have made corresponding corrections/additions to the ms (marked in yellow). Also, supplementary material of Chemtax in terms of input and output ratios is added. Our specific responses are as follows:

R1: Line 12 and entire document: What is the rationale in using the class name for prymnesiophyceae, while all other groups were kept unspecific? Please harmonies all phytoplankton groups throughout the manuscript to prymnesiophytes.

Authors: Yes, the referee is right. We should keep all phytoplankton groups unspecific.

[Figure]

Corresponding changes from prymnesiophyceae to prymnesiophytes have been made throughout the ms.

R2: Line 101 how long after sampling was the filtration done?

Authors: Sampling of filtration was immediately started as soon as returning to the lab and this process usually took no more than one hour. Therefore, the entire time would be around 3 hours (2 h returning journey + 1 h filtration process).

R3: Line 116: Please give more details about the CHEMTAX® approach, did you use different depths, years etc. Please give input and output ratios for the data set.

Authors: All the phytoplankton pigments are dealt with the same input ratio matrix and we have added this information in the ms. Input and output ratios are given in Table S1 as supplementary material.

R4: Line 283 ff and Line 319ff Discussion about 4.3.1 The Major Baltic Inflow events and 4.3.2 Low oxygen events: The paper would benefit, if the two events would be better introduced at the beginning of each chapter including previous literature about associated findings with these type of events. This would be helpful to understand the rationale behind the expectation of changes also for the sulphur compounds.

Authors: We added text as well as relevant references to the ms at the beginning of sections 4.3.1 (The Major Baltic Inflow events) and 4.3.2 (Low oxygen events) in order to introduce these events.

R5: Please check the references for consistency

Authors: We have examined and corrected every reference in the ms to keep the consistency.

Please also note the supplement to this comment:
https://bg.copernicus.org/preprints/bg-2020-431/bg-2020-431-AC2-supplement.pdf

[Figure]

**Supplement:**

**Table S1. Input and output ratios of marker pigments to Chl *a* for the selected phytoplankton groups. Peri, 19'-but, Fuco, 19'-hex, Ddx, Allo, Zea and Chl *b* represent peridinin, 19'-butanoyloxyfucoxanthin, fucoxanthin, 19'-hexanoyloxy fucoxanthin, diadinoxanthin, alloxanthin, zeaxanthin and chlorophyll *b*, respectively. Definition of phytoplankton groups abbreviations is shown in Fig. 5.**

|  | Peri | 19'-but | Fuco | 19'-hex | Ddx | Allo | Zea | Chl *b* | Chl *a* |
|---|---|---|---|---|---|---|---|---|---|
| **Input ratios** | | | | | | | | | |
| Chloro | 0 | 0 | 0 | 0 | 0 | 0 | 0.031 | 0.283 | 1 |
| Dino | 0.547 | 0 | 0 | 0 | 0.247 | 0 | 0 | 0 | 1 |
| Diat | 0 | 0 | 0.81 | 0 | 0.318 | 0 | 0 | 0 | 1 |
| Crypto | 0 | 0 | 0 | 0 | 0 | 0.354 | 0 | 0 | 1 |
| Cyano | 0 | 0 | 0 | 0 | 0 | 0 | 1.337 | 0 | 1 |
| Chryso | 0 | 1.563 | 0.974 | 0 | 0.857 | 0 | 0 | 0 | 1 |
| Prym | 0 | 0.023 | 0.304 | 0.27 | 0.113 | 0 | 0 | 0 | 1 |
| **Output ratios** | | | | | | | | | |
| Chloro | 0 | 0 | 0 | 0 | 0 | 0 | 0.031 | 0.283 | 1 |
| Dino | 0.852 | 0 | 0 | 0 | 0.043 | 0 | 0 | 0 | 1 |
| Diat | 0 | 0 | 0.81 | 0 | 0.044 | 0 | 0 | 0 | 1 |
| Crypto | 0 | 0 | 0 | 0 | 0 | 0.354 | 0 | 0 | 1 |
| Cyano | 0 | 0 | 0 | 0 | 0 | 0 | 1.337 | 0 | 1 |
| Chryso | 0 | 1.563 | 0.974 | 0 | 0.857 | 0 | 0 | 0 | 1 |
| Prym | 0 | 0.023 | 0.304 | 0.27 | 0.113 | 0 | 0 | 0 | 1 |